# Machine learning approaches for predicting the link of the global trade network of liquefied natural gas

**Pei Zhao**[1], **Hao Song**[2*], **Guang Ling**[3]

**1** Department of Educational Information Technology, Beijing Union Univeristy, Beijing, China, **2** Guangxi Key Laboratory of Culture and Tourism Smart Technology, Guilin Tourism University, Guilin, China, **3** School of Science, Wuhan University of Technology, Wuhan, China

* 3001200117@email.cugb.edu.cn

## Abstract

With the rising geopolitical tensions, predicting future trade partners has become a critical topic for the global community. Liquefied natural gas (LNG), recognized as the cleanest burning hydrocarbon, plays a significant role in the transition to a cleaner energy future. As international trade in LNG becomes increasingly volatile, it is essential to assist governments in identifying potential trade partners and analyzing the trade network. Traditionally, forecasts of future mineral and energy resource trade networks have relied on similarity indicators (e.g., CN, AA). This study employs complex network theory to illustrate the characteristics of nodes and edges, as well as the evolution of global LNG trade networks from 2001 to 2020. Utilizing node and edge data from these networks, this research applies machine learning algorithms to predict future links based on local and global similarity-based indices (e.g., CN, JA, PA). The findings indicate that random forest and decision tree algorithms, when used with local similarity-based indices, demonstrate strong predictive performance. The reliability of these algorithms is validated through the Receiver Operating Characteristic Curve (ROC). Additionally, a graph attention network model is developed to predict potential links using edge and motif data. The results indicate robust predictive performance. This study demonstrates that machine learning algorithms—specifically random forest and decision tree—outperform in predicting links within the global LNG trade network based on local information proximity, while the graph attention network, a deep learning model, exhibits stable optimization and effective feature learning. These findings suggest that machine learning approaches hold significant promise for mineral trade network analysis.

## Introduction

Liquefied Natural Gas (LNG) is a significant clean fossil energy source. In light of exacerbating climate risks and evolving energy policies, LNG has assumed an

**Data availability statement:** The data can be downloaded from the US Comtrade database, https://comtradeplus.un.org/.

**Funding:** Sichuan Oil and Gas Development Research Center 2024 General Projects (2024SY004). Key Research Project of China Inorganic Salt Industry Association for 2025: Vulnerability Assessment and Security Optimization of Global Boron Resource Supply and Demand Network (NO. WJYZXYJ25001). This research was funded by National Natural Science Foundation of China grant number 42162028.

**Competing interests:** The authors have declared that no competing interests exist.

increasingly vital role in global energy consumption, with demand growing steadily worldwide. Although global LNG reserves are substantial, they are characterized by uneven spatial distribution [1]. Consequently, the global and regional LNG trade network has experienced rapid growth, diversification, and enhanced flexibility in LNG cargo movements [2]. According to statistics from the International Energy Agency, the global LNG trade volume has increased nearly fourfold, rising from 144 billion cubic meters in 2000–549 billion cubic meters in 2023. This LNG trade network is dynamic and is profoundly influenced by geographical and economic conditions, including political factors, conflicts, GDP fluctuations, and unforeseen events [3]. For instance, the Russia-Ukraine conflict has resulted in increased LNG imports in Europe and altered the dynamics of the LNG trade network, while the outbreak of COVID-19 severely impacted LNG trade during the second and third quarters of 2020. To ensure the security of LNG trade for governments, it is essential to comprehend the impacts of this network and to anticipate potential changes in its structure.

As real-world complex systems increasingly utilize network representations and research, the link prediction problem has emerged as a significant topic in network analysis. Various approaches have been proposed to predict the links within networks. In link prediction analysis, two primary categories of measures are identified: graph-based measures and content-based measures. Graph-based approaches assess potential links based on the topological structure of the network, while content-based measures leverage node or edge information to forecast future links. Graph-based measures are the most fundamental and straightforward approach, social networks, such as trade networks, exhibit complex and dynamic contexts. To enhance prediction performance, researchers have developed algorithms that integrate both more high-order graph-based measures. Motif is defined as the small building blocks of networks, and they are used to statistically overrepresented subgraphs that could reflect the local topology of a network. Some researchers pointed out that motif, a kind of modularity, can be used to forecast the potential links in complex networks [4,5].

In the complex network of global liquefied natural gas (LNG) trade, the connections between countries are influenced by both the topological structure of these networks and various economic and geographic factors. These influences interact with one another and are represented as topological data within the trade network. Previous studies have predicted potential links within the global LNG trade network using graph-based measures. For instance, Jin [6] employed indices based on local information proximity (such as CN, AA, and RA), path proximity (LP), random walk (ACT), and centrality. Additionally, Filimonova et al. [7] analyzed the trade network utilizing graph-based measures, specifically the preferential attachment index. However, it is evident that previous studies on link prediction have lacked high-order graph-based measures. Unsupervised learning methods are used to sort the potential links. It is of value to forecast future links with various indicators with supervised learning algorithms.

To enhance the predictive performance of graph-based measure indicators, supervised machine learning techniques are employed. First, LNGTN is characterized

using complex network theory. Next, this paper predicts potential links utilizing similarity-based indices through machine learning classification models, such as Random Forest and Logistic Regression. Finally, by integrating network edge and motif information, a model based on Graph Attention Networks is developed to deep learn and forecast potential links within the global LNG trade. The notations in this paper is in Table 1.

## 2. Related studies

### 2.1. Global LNG trade network reesarch in complex network theory

Recently, in response to climate change and environmental pollution, an increasing number of countries have implemented new energy policies aimed at exploring clean and low-carbon energy sources. Consequently, the consumption of natural gas has steadily risen. However, due to the uneven distribution of liquefied natural gas (LNG) and variations in supply and demand across different seasons, its trade network has become a significant topic of research in state security and government policy-making [6]. The impacts on global LNG trade encompass gas demand and geopolitical issues [8]. Chen et al. [9] analyzed LNG trade data from 2005 to 2014 to examine trade competition patterns, identifying Qatar, Australia, Malaysia, Algeria, and Nigeria as members of the top ten most competitive countries for LNG exports. They also noted that factors such as economic conditions play crucial roles in LNG trade. Maritime transport is essential for LNG trade, with maritime LNG trade experiencing growth [10]. From the perspective of maritime transportation, Peng et al. [11] utilized global liquefied natural gas trade data from 2013 to 2017 and discovered that Singapore, Ras Laffan, and Khawr Fakkan were the most significant trading hubs in the LNG transportation network. They identified three closely linked trading zones, including a zone that encompasses several ports in the Middle East, Australia, Singapore, East Asia, and Southeast Asia.

In global trade analysis across various domains, the complex network approach is employed to construct network models. This approach utilizes a highly interconnected graph in which a vertex represents an item (e.g., web page, country, person, or paper), and an edge signifies some form of association between the corresponding items (e.g., a trade connection between countries or clients) [12]. Key research areas include performance evolution, structural stability and optimization, cascading failures, link prediction, and robustness, among others [13].

### 2.2. Link prediction in complex networks

**2.2.1. link prediction indicators.** Link prediction refers to the process of identifying absent links in static networks or forecasting the probability of a potential connection between two nodes in an existing dynamic network [14,15]. In the context of trade network analysis, link prediction can be employed to identify prospective trade partners.

Link prediction has emerged as a rapidly growing research area in physics, computer science, and various other domains [14]. There are two primary categories of link prediction measures: graph-based measures and content-based measures. Graph-based measures, also known as topological-information-based link prediction, analyze and predict links

**Table 1. The list of notations in this paper.**

| Notation | Description |
| --- | --- |
| $V$ | The set of nodes in the global LNG trade network |
| $E$ | The set of edges in the global LNG trade network |
| $G = (V, E)$ | The graph for the global LNG trade network |
| $\Gamma_x$ | The set of neighbors of the node x |
| LNGTN | Global trade network of LNG |
| ReLU | Rectified linear unit |
| GAN | Graph Attention Network |

based on the topological structure of networks. Key indices in this category include Common Neighbors (CN), Adamic-Adar (AA), Katz, Resource Allocation, Preferential Attachment, and Jaccard, among others. In contrast, content-based measures, or node-information-based link prediction, utilize the attributes of the vertices and edges within the networks [15].

In addition to common feature indicators, other network topological features, such as modularity and motifs, are considered for predicting future links [4,16]. Motifs can reflect the local topology of a network [17]. First introduced in "Science" in 2002, motifs enable the examination of the local structure of complex networks and facilitate a comprehensive analysis of their topological properties and implicit correlation features [18]. Consequently, motif-based analysis has emerged as a research hotspot in the in-depth study of large-scale complex networks. Among the various types of motifs (Fig 1), triads (three-node motifs) are particularly utilized. Farshad & Mohammad [19] proposed that a motif is constructed from each pair of nodes and one of their common neighbors. There are 13 types of three-node motifs, categorized into 13 morphism classes of triads (as shown in Fig 1) [17]. The P-value and Z-score of motif recurrence are calculated within a vast pool of randomly generated graphs.

In addition to these indicators, time series analyses in dynamic networks are employed to evaluate potential links. Huang and Lin [20] proposed a time series (TS) analysis approach that integrates static scoring methods to predict network links. Da Silva Soares and Bastos Cavalcante Predencio [21] introduced both unsupervised and supervised link prediction models within TS approaches, demonstrating satisfactory results.

**2.2.2. link prediction methods.** The indicators for all node pairs are employed to perform link prediction using either an unsupervised or a supervised method. The unsupervised method involves sorting links in descending order based on their scores, while the supervised methods approach the prediction problem as a binary classification task [22]. Historically, many studies have utilized unsupervised methods to evaluate proximity scores. However, with advancements in big data analysis and machine learning, an increasing number of papers are adopting supervised methods, incorporating various features with differing coefficients derived from training data [22].

With the rise of large-scale, complex, and dynamic networks, machine learning algorithms have emerged as a major strategy for link prediction, significantly enhancing precision. The abundance of big data in social networks presents opportunities for link prediction algorithms to perform calculations, data processing, and automated prediction tasks. This approach has proven to be one of the most effective methods in the field. However, it requires additional information to effectively model link prediction, such as various factors influencing link formation. Mohan et al. [23] utilized community structure information within the Bulk Synchronous Parallel programming model as a graph algorithm for link prediction in large networks. Similarly, Li et al. [24] introduced a deep learning framework incorporating temporal restricted Boltzmann machines and gradient boosting decision trees for large network analysis.

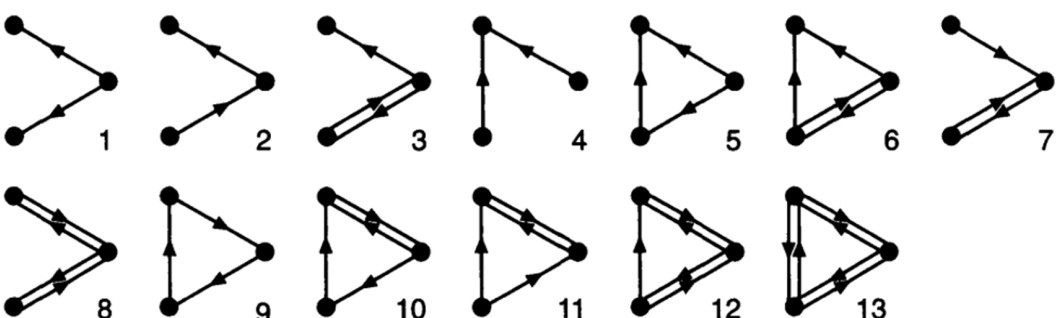

**Fig 1. 13 morphism classes of triads (Milo, et al. 2002).**

**2.2.3. Supervised learning in link prediction.** Recently, the study of network theory has increasingly focused on link prediction as a prominent research area. State-of-the-art machine learning techniques are being applied across various domains, including social networks, trade networks, and biological networks, to predict potential links within complex networks. Generally, there are two primary categories of approaches for link prediction: unsupervised approaches and supervised approaches. Unsupervised methods compute scores for connections between nodes based on topological information, such as k-means clustering, while supervised approaches aim to predict new links by learning a parameter vector through techniques like regression and classification [12]. Supervised learning approaches can be further categorized into feature-based methods, graph regularization-based methods, latent class-based methods, and latent feature-based methods [12]. The integration of unsupervised and supervised learning approaches has been shown to enhance the accuracy and efficacy of link prediction, leading researchers to refer to this combined method as 'semi-supervised learning.' This approach utilizes a substantial amount of unlabeled data alongside a limited set of labeled data to perform learning tasks [25,26]. Several methods exist within the semi-supervised framework for link prediction. For instance, Kashima et al. [27] employed the principle of label propagation, which posits that two similar nodes are likely to share the same label and link. Additionally, matrix factorization techniques, including singular value decomposition (SVD) and principal component analysis (PCA), have been applied to predict links.

Among several link prediction algorithms, embedding-based algorithms utilize learned vectorized features to measure the similarity between two vertices. In contrast to traditional similarity-based algorithms in link prediction, such as CN and AA, embedding-based algorithms can effectively address higher-order structures, weak ties, and complex configurations present in real-world complex networks [5]. Common algorithms for deep learning the vertex features of a network to identify potential links include SDNE, Node2vec, HONE, and DeepDL. Recently, motifs have been employed to learn vectorized features. Wang et al. [5] introduced a learning algorithm called MODEL, which predicts links using motifs. This approach is commonly applied in biological networks and social networks, such as trade networks. For instance, Zhang et al. [28] integrated motif analysis to predict potential international trade relations involving nine crucial minerals, which serve as primary raw materials for lithium-ion batteries.

**2.2.4. Deep learning with GATs in link prediction.** Currently, some researchers are utilizing Graph Neural Networks (GNNs) to predict potential links. Graph Attention Networks (GATs) and Graph Convolutional Neural Networks (GCNs) are two state-of-the-art architectures employed in network analysis [29]. For instance, Cheng, Wang, and Tan [30] proposed a novel link prediction method for a Chinese financial event knowledge graph based on Graph Attention Networks and Convolutional Neural Networks. Compared to the popular research topic of GCNs in complex network data mining tasks, GATs are capable of understanding high-order structural features of networks and have demonstrated strong performance in node classification tasks. Sheng et al. [31] utilized high-order structures—motifs—in GATs and constructed the MGAT and MGATv2 models.

GATs use attention mechanisms to weigh the importance of nodes' neighbors, demonstrating flexibility and power in representation learning [32]. In GCNs, the computation process is formalized: $h_i' = f(h_i, AGGREGATE(\{h_i | j \in N_i\}))$ . GATs introduce the concept of attention mechanisms, and it could allow the aggregation function to adaptively match different neighboring nodes with corresponding weights. GATs calculate attention scores $e : R^d \times R^d \to R$ for each edge $(i,j)$ to quantify the importance of neighbor $j$ to node $i$, and the computation process can be described as following: $e_{ij} = \sigma(a^T \cdot [Wh_i || Wh_j])$ where $\sigma$ represents the non-linear activation function ReLU. The core working principle of GATs is to calculate the relationship between nodes through the attention mechanism. GATs could receive more information based on different features and models from every attention heads. They utilize attention mechanisms to learn discriminative features, thus reducing the over-smoothing problem [33].

## 2.3. Link prediction in energy and mineral resource trade network researches

Exploring international trade relations constitutes the primary research focus in global trade. Various models exist to anticipate potential connections within complex networks. For instance, the gravity model is predominantly utilized in the

energy and resource trade network; however, its predictions regarding potential trade relations are not optimal [6]. Some researchers have employed complex network approaches to analyze trade networks and identify potential links, constructing both undirected and directed networks. They subsequently predicted these links based on the topological features of the networks [7,34].

Energy and mineral resources play a crucial role in emerging strategic industries at the national level. However, given the dynamic and complex nature of global resource trade networks, it is essential to understand their evolution and future trends. The identification of hidden or potential trade routes is a key topic in the analysis of trade network trends. Several researchers have employed link prediction methods to anticipate potential trade routes within the framework of complex network theory. Zhu et al. [35] utilized four graph-based measures (CN, AA, RA, and PA algorithms) to predict international trade links for boron ore, finding that competition among Latin America, Europe, and Africa is likely to intensify. Wei, Xie, and Zhou [36] assessed the performance of international oil trade networks. Some researchers have explored to use link prediction approach to find the potential trade links based on the topological attributes of countries [37]. In the context of link prediction for LNG, potential links are typically anticipated based on topological features [8]. However, most studies have focused on unsupervised link prediction models, and there is a notable lack of research utilizing supervised learning algorithms with high-order trade network topological data. Literature review table is Table 2.

## 3. Data and method

### 3.1. Description of liquified natural gas trade data

Utilizing complex network theory, we selected the annual international import trade data for liquefied natural gas (LNG) from the UN COMTRADE database (available at http://comtrade.un.org/) for the years 2001–2020, specifically using the HS code 271,111. This dataset encompasses information on global LNG trade flows among all participating countries and regions. The trade volume is quantified in kilograms. To enhance the efficacy of our algorithms, we refined the dataset by excluding trading countries with zero imports.

The global LNG trade network (LNGTN) can be represented as a complex network. We employed an undirected weighted graph. In the graph $G_{trade} = (V, E, t)$. In this model, $V = \{v_1, v_2, \ldots, v_n\}$ is the set of all import and export countries (nodes) in the network, and $E = \{e_1, e_2, \ldots e_{ij}\}$ is the set of all trade relationships, $e_{ij}$ represents the link between country (node) i and j.

### 3.2. Research steps

This study presents a comprehensive machine learning framework for predicting LNG trade network evolution through systematic link prediction analysis. The methodology unfolds in three key phases: First, we construct predictive features by computing both local (CN, AA, PA) and global (Katz) network similarity indices, while implementing a temporal labeling system to identify emerging links. Second, we conduct comparative modeling using five machine learning classifiers (Logistic Regression, Random Forest, Decision Tree, SVM) alongside a Graph Attention Network, with detailed hyperparameter specifications for each algorithm. Third, we establish a rigorous evaluation protocol employing stratified 80%−20% data splitting, cross-validation, and multiple performance metrics (F1-score, AUC-ROC) to ensure robust assessment. This integrated approach uniquely combines network science principles with advanced machine learning techniques, offering both methodological rigor through its systematic design and practical utility through its comparative algorithm evaluation, ultimately providing a reproducible framework for analyzing dynamic trade network evolution.

### 3.3. Features extraction in link prediction

As the trade network -LNGTN- is constructed based on the trade volume and value of each country, this paper employs supervised machine learning approaches to determine the probability of establishing a link between two isolated nodes in the future.

**Table 2. literature review table.**

| Author(s) | Year | Research Focus | Methods/Models | Key Findings | Limitations | Your Research Innovation |
|---|---|---|---|---|---|---|
| Chen et al. | 2016 | Global LNG trade competition analysis | Network analysis of trade data (2005–2014), competitive country ranking | Identified top LNG exporters (Qatar, Australia, etc.) and role of economic factors | Static analysis; no link prediction; | Dynamic network modeling + link prediction to forecast evolving trade relationships |
| Peng et al. | 2021 | Maritime LNG trade network structure | Complex network analysis of port-level trade data (2013–2017) | Identified key hubs (Singapore, Ras Laffan) and regional trade clusters | Focus on static topology; no prediction of potential trade links or new routes | Incorporate link prediction to identify unexploited maritime trade opportunities |
| Huang & Lin | 2009 | Time-series-based link prediction in dynamic networks | Static scoring + time-series (TS) analysis | TS improves prediction accuracy in temporal networks | Limited to simple topological features; no deep learning integration | Deep learning with temporal data (e.g., GATs) to model time-varying LNG trade patterns |
| Wang et al. | 2020 | Motif-based deep learning for link prediction | MODEL algorithm (motif features + deep neural networks) | Motifs improve prediction of high-order network structures (e.g., triads) | Applied primarily to social/biological networks; not tested on energy trade networks | Motif analysis in LNG trade networks to capture hidden topological dependencies |
| Sheng et al. | 2024 | Graph Attention Networks (GATs) with motifs (MGAT) | Motif-based GATs for high-order structure learning | MGAT outperforms traditional GCNs in node classification tasks | Not applied to directed trade networks (e.g., LNG export-import dynamics) | Directed motif-GAT model for asymmetric LNG trade relationships |
| Zhu et al. | 2023 | Boron ore trade prediction using graph-based measures | Unsupervised link prediction (CN, AA, RA, PA algorithms) | Predicted intensified competition in Latin America/Europe/Africa for boron ore | Relied on unsupervised methods; lacked integration of economic/geopolitical features | Supervised learning + multi-source features (e.g., tariffs, demand) for LNG trade prediction |
| Jin et al. | 2022 | LNG potential trade relations using improved link prediction | Complex network analysis with gravity model adjustments | Identified underdeveloped trade links but used traditional similarity metrics | Limited to static topological metrics; no deep learning or motif-based features | Deep motif-GAT framework combining topological, temporal, and contextual data |
| Hou et al. | 2024 | Spatiotemporal evolution of LNG trade networks | Multilevel network analysis (spatial-temporal dynamics) | Identified spatial-temporal patterns in LNG trade flows | Focus on descriptive analysis; no predictive modeling | Integrate spatiotemporal dynamics into link prediction models |
| Filimonova et al. | 2022 | Transformation of LNG trade routes | Network analysis of new trade routes and geopolitical impacts | Identified emerging trade routes (e.g., Russia-Asia) and their stability | Focus on static route mapping; no link prediction | Predictive modeling of route stability under geopolitical changes |
| Dai et al. | 2017 | Link prediction based on modularity | Modularity-based similarity metrics | Modularity improves prediction of community-level interactions | Limited to undirected networks; no dynamic analysis | Directed modularity analysis for asymmetric LNG trade communities |
| Pecli et al. | 2018 | Supervised learning for link prediction in trade networks | Comparative study of supervised algorithms (e.g., SVM, Random Forest) | Supervised methods outperform unsupervised ones when contextual features are available | Limited to social networks; not tested on energy trade | Supervised learning with domain-specific features (e.g., LNG price volatility) |
| Guan et al. | 2016 | Crude oil trade prediction using link prediction | Graph-based measures (CN, AA, RA) applied to directed trade networks | Predicted potential crude oil trade links with high accuracy | Limited to crude oil; not validated on LNG trade | Adaptation to LNG trade with directed network modeling |

**3.3.1. Graph-based measures features.** The network's topological information is utilized to calculate similarity-based features for the supervised learning model. This paper employs graph-based measures to derive these features. In the context of graph-based measures, the indices are categorized into two groups: local and global similarity-based indices [38].

In local similarity indexes, there are three approaches—information-based, path-based, and random-walk-based approaches [16]. The local information-based approaches include the following indices::

(1) The Common Neighbors Index (CN) posits that the likelihood of an edge forming between two nodes increases when they share a significant number of neighboring nodes [39]. It is based on the proximity of the nodes in the network. This method is defined as:

$$S_{CN}(x, y) = |\Gamma_x \cap \Gamma_y| \tag{1}$$

Where $\Gamma(x)$ and $\Gamma(y)$ are the neighbor sets of node x and y, respectively. However, it does not consider the influence of the node degree.

(2) The Jaccard Coefficient (JC) index is influenced by the degree of the network nodes; specifically, if the proportion of common neighbors to the total number of neighbors is higher for a pair of nodes, their JC index will also reflect elevated values. A related measure is the Sorensen index (SO), which quantifies the relative size of the intersection of neighbor sets [32]. They are defined in the following equations:

$$S_{JC}(x, y) = \frac{\{\Gamma_x \cap \Gamma_y\}}{|\Gamma_x \cup \Gamma_y|} \tag{2}$$

$$S_{SO}(x, y) = \frac{2|\Gamma_x \cap \Gamma_y|}{|\Gamma_x| + |\Gamma_y|} \tag{3}$$

Where $\Gamma_x$ and $\Gamma_y$ stand for all neighboring vertices of node x and y.

(3) The Adamic-Adar Index (AA) is proposed as a measure of similarity between two entities based on their shared features within social networks [40]. It is analogous to the CN index; however, this method posits that in a network, nodes with a large number of neighbors are less significant in predicting a connection between two nodes compared to those with fewer neighbors. Each node is assigned a weight according to the degree of its common neighbors. Consequently, this index indicates that if node pairs share fewer common neighbors, they are weighted more heavily [28]. It could be described as (Equation 3 and 4 are used in unweighted and weighted network):

$$S_{AA}(x, y) = \sum_{z \in \Gamma_x \Gamma_y} \frac{1}{\log |\Gamma_z|} \tag{4}$$

$$S_{AA}(x, y) = \sum_{z \in \Gamma_{out}(x) \cap \Gamma_{in}(y)} \frac{w_{xz} + w_{zy}}{\log |1 + s_z|} \tag{5}$$

In the Equation 3, $\Gamma_z$ is the set of nodes that are adjacent to z. Z is the node set which are the common neighbors of nodes x and y. while in the Equation 4, $s_z$ represents the sum of the trade volume of z. It could be described $s_z = \sum_{i \in \Gamma_z} w_{zi}$ , where country i has trade relationship with country z.

(4) The Resource Allocation index (RA) posits that for a pair of nodes, x and y, which are not directly connected, the similarity between them can be defined by the amount of resource that node y receives from node x. This index utilizes the degrees of the nodes to quantify the contributions of their common neighbors [28]. The equations of the unweighted and weighted networks are described as:

$$S_{RA}(x, y) = \sum_{z \in \Gamma_x \cap \Gamma_y} \frac{1}{k_z} \tag{6}$$

$$S_{RA}(x, y) = \sum_{z \in \Gamma_{out}(x) \cap \Gamma_{in}(y)} \frac{w_{xz} + w_{zy}}{s_z} \tag{7}$$

In the Equation 5, the nodes in the set of z are the common neighbors of nodes x and y as the AA index. $k_z$ is the degree of node set z. $s_z$ in the Equation 6 has the same meanings as that in the Equation 4.

(5) The Preferential Attachment Index (PA) demonstrates that nodes with higher degrees are more likely to be interconnected. This index is commonly applied in scale-free networks, where the degree distribution of nodes adheres to a power law. The similarity can be quantified using the following equations: Equation 7 is applicable to unweighted networks, while Equation 8 pertains to weighted networks.

$$S_{PA}(x, y) = |\Gamma_x| \times |\Gamma_y| \tag{8}$$

$$S_{PA}(x, y) = \sum_{j \in \Gamma_{out}(x)} w_{xj} \times \sum_{q \in \Gamma_{in}(y)} w_{yq} \tag{9}$$

In the Equation 8, j represents the trade partner that imports from country x, and q is the country that exports to country y. Even though PA is easy to understand, its prediction accuracy is usually poor, especially in a global measure.

(6) The Hub Promoted Index (HPI) and Hub Depressed Index (HDI). HPI was proposed by studying modularity in metabolic networks in 2002 [41]. This index attempts to build the connection between nodes and the low-level nodes, not the nodes with same levels [42]. It is defined as. In contrast to HPI, HDI penalized big neighborhoods [32].

$$S_{HPI}(x, y) = \frac{|\Gamma_x \cap \Gamma_y|}{min\{|\Gamma_x|, |\Gamma_y|\}}$$

$$S_{HDI}(x, y) = \frac{|\Gamma_x \cap \Gamma_y|}{max\{|\Gamma_x|, |\Gamma_y|\}}$$

### 3.3.2. Global similarity-based indices.

Besides local similarity indexes, some methods used the total topological information to predict the future links, and they are named global similarity-based indices, like shortest path, Katz centrality index and Random Walk with Restart.

(7) The Shortest Path (SP). Dijkstra's algorithm is an effective way to compute shortest paths $O_{(ev \log v)}$. SP can favor the closest nodes to form the new links in a network [43]. This index is defined as:

$$S_{sp}(x, y) = -|dist(x, y)| \tag{10}$$

(8) The Katz centrality index (KI) is a well-established method within the path-based approach that takes into account all possible paths in a network [44]. The similarity between two nodes is weighted by the sum of the paths that connect them [45]. Shorter paths are assigned greater weight, whereas longer paths receive less weight. Consequently, shorter paths play a more significant role. It is written as the following equation:

$$S_{Katz}(i, j) = \sum_{i=1}^{\infty} \beta^l path_{i,j}^l \tag{11}$$

where $path_{x,y}^l$ is the set of the path of length l between nodes $i$ and $j$, and $\beta^l$ is the decay factor corresponding to the path length l. The parameter $\beta$ is a damping factor with $0 < \beta < 1$.

(9) The PageRank (PR) algorithm was developed by Google co-founders Larry Page and Sergey Brin at Stanford University to measure the importance of a web page within a web graph. Its principle is that if a vertex has many numerous and significant neighbors, it is considered important; furthermore, the more important a vertex is, the more it conveys its importance to its neighbors. This measure is written by the following equation in Google paper:

$$PR_p = (1-d)d\left(\frac{PR(T_1)}{C(T_1)} + \ldots + \frac{PR(T_n)}{C(T_n)}\right)$$

(12)

In this formula, d is the damping factor that is from 0 to 1 (normally 0.85), and C(p)is the number of outgoing links from p web page node. This index could be used to predict the potential link of a network by calculate the similarity between the PR of x and y, especially if x and y are not connected structurally. Charikhi [42] proposed this index with local similarity indexes [46] and found it has a good performance.

(10) Random Walk with Restart (RWR) indices are described as a random walk optimization problem. RWR is a widely used measure of node similarity in networks and can be regarded as an application of the PageRank algorithm in link prediction. This paper employs RWR to predict links within a network. It quantifies the similarity between two nodes (x, y) by the probability of a random walker starting at x and ultimately reaching y [47]. For a given node x, the RWR proximity values from it to other nodes could be defined as the equation:

$$p_x = (1-\alpha)Ap_x + \alpha e_x$$

(13)

Where $p_x \epsilon R^n$ is the proximity vector of node x, with $p_x(v)$ denoting the proximity from x to v. $\alpha$ is a damping factor from 0 to 1, and it denotes the restarting probability in RWR (typically, $\alpha = 0.15$).In the link prediction, RWR could be used to describe the similarity of two nodes (x, y) by the maximum of their RWR values.

$$S_{RWR}(x,y) = Max(RWR_{(x)}, RWR_{(y)})$$

(14)

All indices mentioned utilize the common topological information of a complex network. Regarding the motif feature, several algorithms have been developed, such as triadic similarity and motif-based similarity indices

**3.3.3. Similarity-based indices with motifs.** In addition, high-order topological similarity-based information—motif—was introduced in this paper. Network motifs are the sub-graphs which are defined by particular patterns of interactions between vertices. Several researches proposed motif analysis methods:

(11) Farshad & Mohammad [19] measured the triadic similarity between two nodes through a supervised learning experiment framework. The similarity is estimated as the following equitation:

$$S_{triadic\ similarity}(x,y) = \frac{\sum_{z\in\Gamma_{(x)}\cap\Gamma_{(y)}} \phi_{(x,y,z)}\times\frac{1}{13}}{|\Gamma_{(x)}\cap\Gamma_{(y)}|}$$

(15)

where, $\phi_{(x,y,z)}$ is the number of the with x, y and z as the participant nodes.

(12) Motif-based similarity Index (MS). Li, Wei and Liu [34] stated that if the intersection of the neighbor structures of the node pairs to be predicted is relatively high, these nodes are probably linked. This index is proposed as:

$$S_{MS}(u,v) = \frac{|\Gamma_{(u)}\cap\Gamma_{(v)}|}{|\Gamma_{(u)}\cap\Gamma_{(v)}|+motif_{(u)}+motif_{(v)}}$$

(16)

where motif(u) and motif(v) denote the motif structures containing u and v, respectively. In this paper, the triangular motif structure is used to measure the local correlation between nodes in the network.

In this study, we examine various indicators for link prediction using machine learning algorithms. These indicators include local similarity-based measures (CN, JC, AA, JA, and PA), global similarity-based measures (Katz and RWR), and motif analysis indicators (triadic similarity).

### 3.4. Supervised machine learning algorithms in link prediction

Most baseline predictors in link prediction employ scoring methods. Various algorithms have been developed for link prediction, and machine learning ensemble techniques provide a robust approach to enhancing prediction accuracy [30]. In the context of proximity scores within complex networks, several classical classifiers, such as Decision Trees, k-nearest neighbors, Naïve Bayes, Bayes Net, Support Vector Machine (SVM), and Logistic Regression, have been utilized [48]. Compared to traditional unsupervised learning methods, supervised machine learning techniques are recognized for their superior prediction accuracy; however, they remain constrained by factors such as feature selection, dataset size, dataset imbalance, and data cleaning [48]. In this paper, we employ several machine learning algorithms to empirically investigate link prediction within the LNG trade network, specifically Logistic Regression, Decision Trees, Random Forests, and K Nearest Neighbors, in order to identify the feature set that yields high accuracy. In addition to common proximity scores, motif-based analysis is also utilized to assess the topological features of the network.

In addition to machine learning algorithms, deep learning methods are also employed in link prediction. Talawar, Ashoka, and Nagaraja [49] proposed a reliable communication link prediction-based traffic-aware deep learning routing protocol for mobile ad-hoc networks, utilizing Fuzzy-based Deep Extreme Q-learning. Chen et al. [10] utilized Long Short-Term Memory (LSTM) networks to develop a deep learning framework for dynamic network link prediction. Generative Adversarial Networks (GANs), a typical model of Graph Neural Networks (GNNs), were employed to learn network edge information and motif data to predict future links. According to GANs theory [33], the importance of node j to node i is calculated using the following formulation:

$$a_{ij} = a(W\vec{h_i}, W\vec{h_j})$$

(17)

Where $W$ is a trainable weight matrix and a is a shared attention mechanism. In this study, GANs were utilized to construct a model that deep learns high-order topological features.

The GAT model incorporates two attention mechanisms: a novel self-attention mechanism and multi-head attention. In this model, the novel self-attention mechanism assesses the influence of edges and motifs. To learn the weights of the relationships among neighbors, we utilize edge and motif adjacency matrices. The motif adjacency matrices are computed using the formula provided in Equation 14. The trade weight value is derived to characterize the edge matrices through mean normalization. These two matrices are merged into a single matrix using ReLU and mean normalization functions, as illustrated in Fig 2.

### 3.5. Research progress

(1) In this link prediction research, local similarity indices (e.g., CN, AA, PA, JA), global indices (e.g., Katz, PageRank), as well as edge and motif structures in the $n$ year are considered as features in the machine learning framework for identifying potential links.

(2) As for the label data, in the machine learning method, we aim to discover links that do not currently exist. Specifically, we utilized new trade relationships between two nodes that exhibit high feature scores in the $n$ year. If this relationship is established in the $n+1$ year, it is labeled as 'yes'; otherwise, it is labeled as 'no'. In the deep learning method with GATs model, the label value is the weight value of edge by mean normalization in the $n+1$ year.

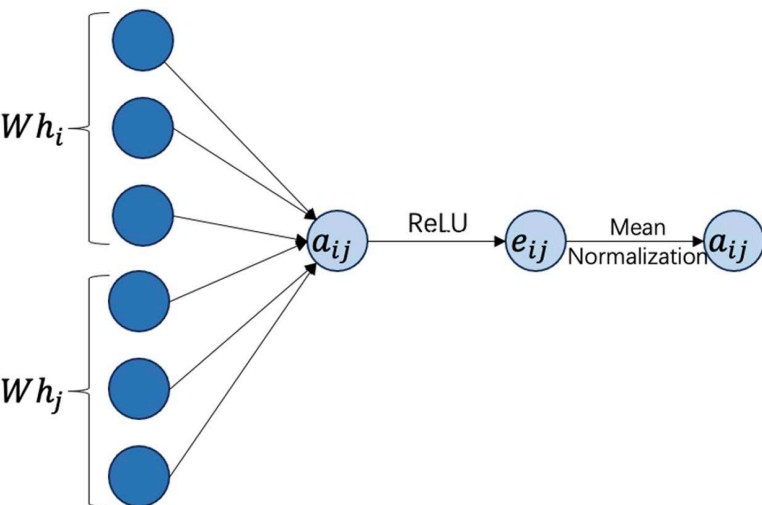

**Fig 2. The model of novel attention mechanism of GATs in predicting the potential links in LNGTN.**

(3) The data is divided into a Training Set and a Test Set. The algorithms randomly select 10% of the trade link data (E) as the test set ($E_{te}$), and the remaining data are used as the training set ($E_{tr}$).

(4) To evaluate the link prediction indices, we employed metrics such as the area under the precision-recall curve (AUC), accuracy, and precision to assess the effectiveness of the link prediction algorithms. A higher AUC score indicates greater accuracy of the corresponding link prediction algorithm. The AUC and precision metrics are calculated using specific equations.

$$AUC = \frac{n' + 0.5n}{n} \tag{18}$$

$$Precision = \frac{L_m}{L} \tag{19}$$

where $n$ is the number of independent comparisons of score of a randomly selected missing link and a non-existed link, $n'$ is the number of occurrences that the missing link has a higher score [50].

In the context of the AUC equation, if all scores are generated randomly, the AUC score is approximately 0.5. A higher AUC score indicates greater accuracy of the corresponding algorithm. The AUC score is defined as the area under the ROC curve (Receiver Operating Characteristic curve). Accuracy is defined as the proportion of correctly predicted samples relative to the total number of samples. In the Precision equation, when predicting the top L potential links, and $L_m$ links among them are correct.

## 4. Results and discussion

### 4.1. Global LNG trade network overall analysis

Using complex network and visual graph methods, we construct a directed global LNG trade network based on UN Comtrade data (Fig 3), with ports as nodes and inter-port LNG trade links as edges. Fig 1 illustrates the trade network chord diagrams for the years 2001, 2006, 2011, 2016, and 2021. These diagrams show that from 2001 to 2016, Japan was the largest importer of LNG from various partners. Additionally, the major exporting countries evolved from Southeast Asian nations (e.g., Malaysia, Indonesia) to those in the Middle East (e.g., Qatar). By 2021, the largest LNG trade globally occurred between the USA and Mexico.

**Fig 3. 2000-2021 Global LNG trade network chords (2000,2006,2011,2016,2021 Year).**

## 4.2. Link Prediction by Supervised Learning

### 4.2.1. Machine Learning with local similarity-based index.

In the annual LNG trade network, this paper selects five local similarity indexes—CN, JA, AA, RA, and PA—as features for predicting potential links. The relationships among these five indexes and their corresponding labels are analyzed, as illustrated in the following Fig. The analysis reveals that among the five similarity-based indexes, the Jaccard coefficient and resource allocation indexes (Fig 4) exhibit a weak relationship with the labels, while the other three indexes show a positive correlation.

In this section, we concentrate solely on topological local similarity-based features derived from machine learning algorithms. To facilitate our learning process, we have compiled all pertinent features from previous years, with the labels corresponding to the subsequent year. For example, the features encompass the Common Neighbors (CN), Adamic-Adar (AA), and Preferential Attachment (PA) scores from 2001, while the labels indicate whether a new relationship was established in 2002. The machine learning process integrates local similarity-based scores, including Common Neighbors, the Adamic-Adar index, the Resource Allocation index, and Preferential Attachment.

The process for link prediction in LNGTN using network similarity-based indices is illustrated in Fig 5 below. First, LNG trade networks are represented through the lens of complex network theory. Second, topological local and global

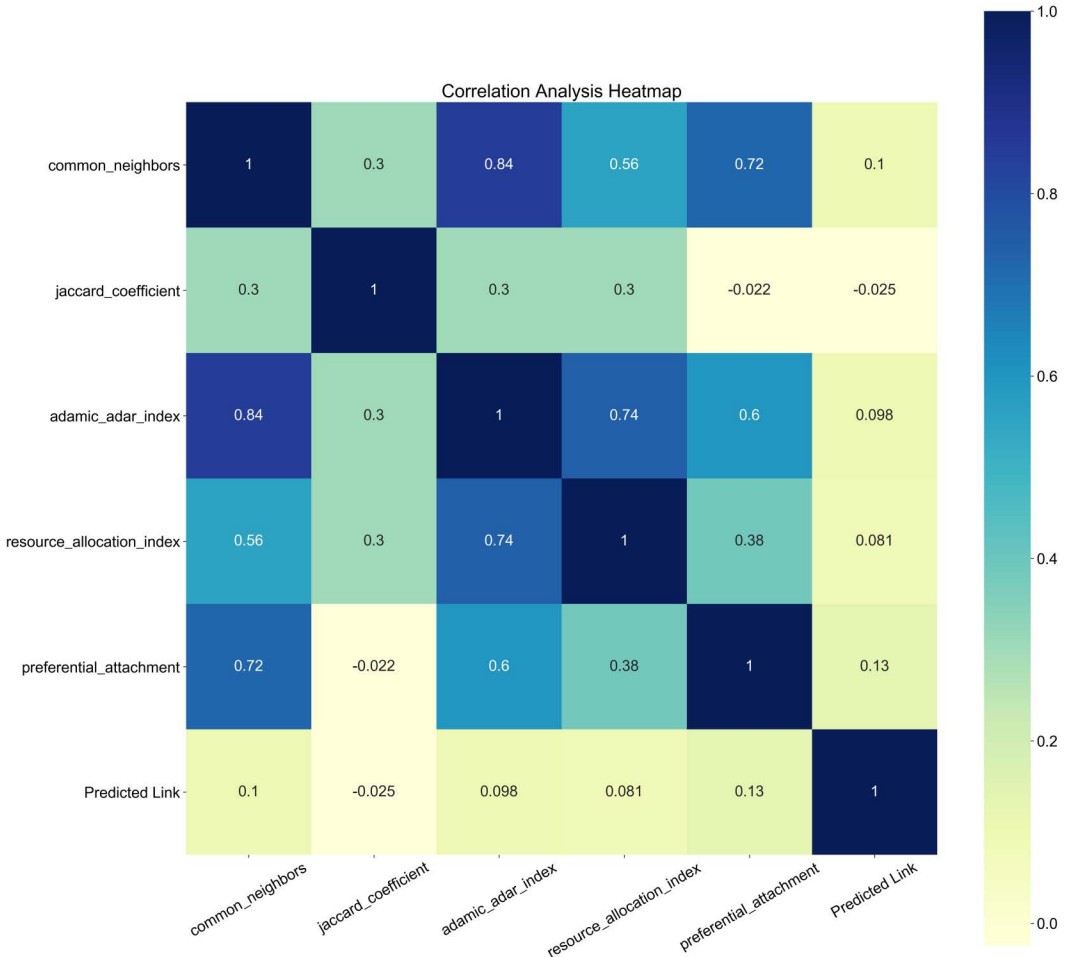

**Fig 4. The Jaccard coefficient and resource allocation indexes.**

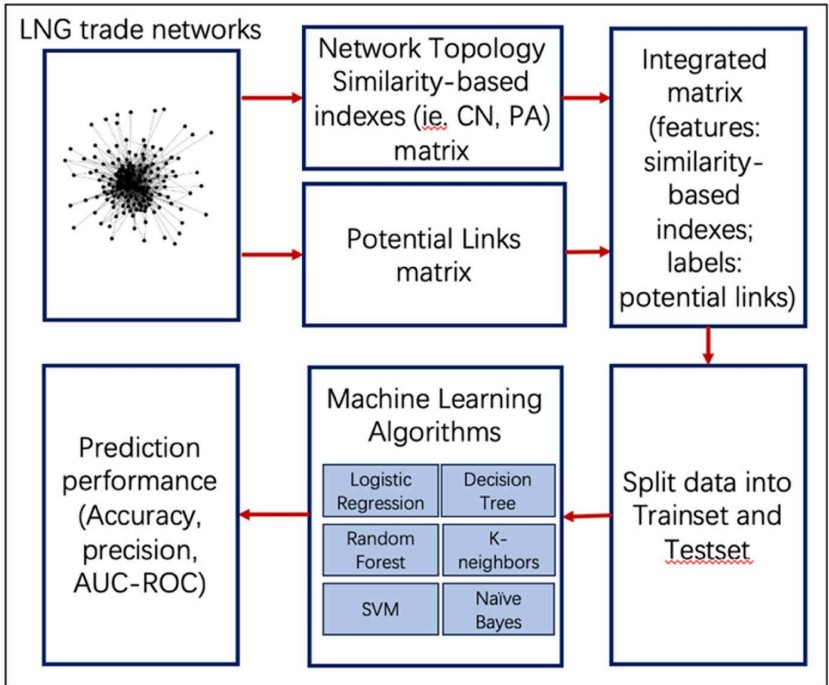

**Fig 5. The overview of process of link by machine learning with local- and global- similarity-based scores.**

similarity-based indices, such as CN, PA, and Katz, are calculated and combined into a matrix that serves as the feature set for potential links. Labels are manually assigned to the matrix, indicating whether potential links will be established in the following year. Third, the matrix is divided into a training set and a test set (Table 3). The data is then trained using machine learning models, including Logistic Regression and Random Forest. Finally, metrics such as accuracy, precision, and ROC curves are employed to evaluate the predictive performance of the network.

```
Algorithm 1: Link Prediction Model with local- and global- similarity-based scores
Input: G=(V, E) where V represents Nodes and E represents links in the network
Output: Predict new link performance
1.Preprocess data by clean nan value
2.Calculate local- and global- based indexes of two nodes as features, and link established or not
as labels.
3.Construct Train Dataset and Test Dataset
4.Perform the link
5.Perform link prediction
```

To predict trade links between countries based on topological features, we utilize and compare various supervised machine learning classification and regression approaches, including Decision Trees, k-nearest neighbors, Support Vector Machines (SVM), Naïve Bayes, Bayesian Networks, and logistic regression, among others. The predictive performances

**Table 3. Division of the LNG trade network into training/test and prediction sets.**

| Features | Labels | Set |
|---|---|---|
| 2001-2020 | 2002-2021 | Training/Test Set |
| 2021 | 2022 | Prediction Set |

of these six algorithms are presented and compared in Table 4. The results indicate that the Decision Tree and Random Forest algorithms demonstrate significantly better performance than the other methods.

As shown in Fig 6, all six models demonstrate strong performance in link prediction through supervised machine learning. Among these, the Decision Tree and Random Forest algorithms exhibit the highest levels of accuracy and precision. According to the ROC curves presented in Fig 2, the AUC score for the Random Forest algorithm is superior to that of the Decision Tree, with values of 0.93 and 0.92, respectively. A comparison of the six algorithms in learning similarity-based indexes indicates that the Random Forest and Decision Tree models are recommended for predicting potential links within global LNG trade networks. By combining the AUC scores and ROC curves, it is evident that the Random Forest algorithm outperforms the Decision Tree algorithm.

The performance of most machine learning algorithms can be enhanced through careful hyperparameter optimization. This study employs hyperparameter tuning to improve the predictive accuracy of both Random Forest and Decision Tree models. For the Random Forest model, key parameters such as node size, the number of trees, and the splitting rule

**Table 4. Link prediction performance of different machine learning model with local similarity-based scores.**

|  |  | Logistic Regression Algorithm | Decision Tree Algorithm | Random Forest Algorithm | K-nearest Neighbors Algorithm | SVM Algorithm | Naïve Bayes Algorithm |
|---|---|---|---|---|---|---|---|
| Accuracy |  | 0.94 | 0.97 | 0.97 | 0.90 | 0.94 | 0.88 |
| Precision | 0 | 0.94 | 0.98 | 0.98 | 0.94 | 0.94 | 0.94 |
|  | 1 | 0.00 | 0.87 | 0.86 | 0.16 | 0.00 | 0.16 |
| F1-score | 0 | 0.97 | 0.98 | 0.99 | 0.95 | 0.97 | 0.94 |
|  | 1 | 0.00 | 0.77 | 0.78 | 0.15 | 0.00 | 0.15 |
| AUC |  | 0.70 | 0.92 | 0.93 | 0.57 | 0.84 | 0.66 |

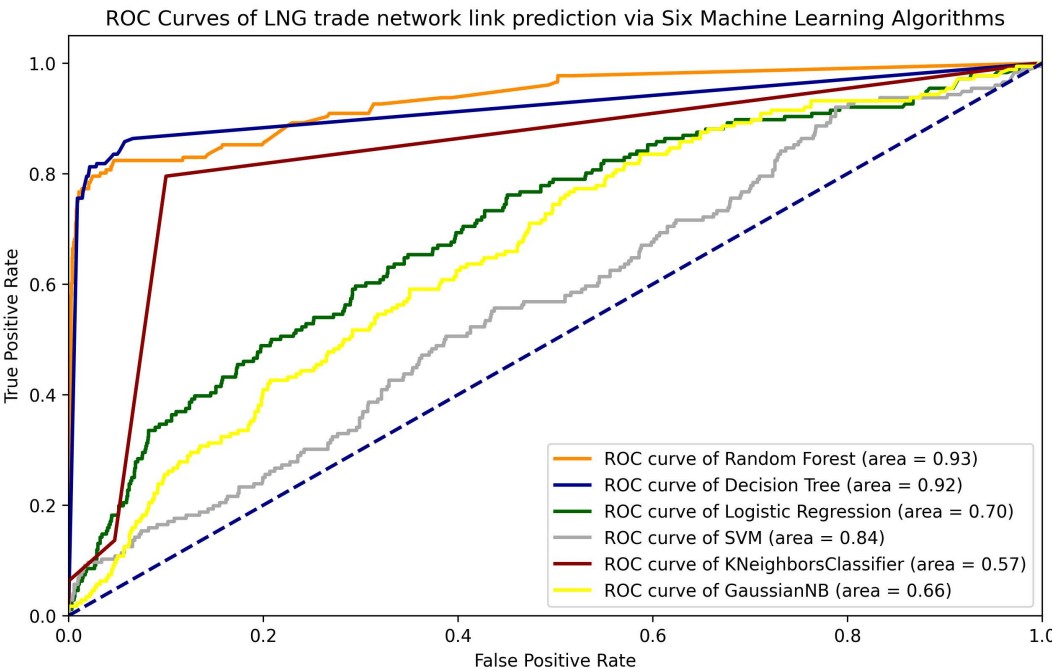

**Fig 6. AUC-ROC curves of LNG trade network link prediction via Six Machine Learning Algorithms.**

significantly influence performance [51]. Similarly, in the Decision Tree model, critical hyperparameters include pruning confidence, the minimum number of instances per leaf, binary splits, and subtree raising [52]. Using the GridSearchCV method for systematic parameter optimization, the study found that hyperparameter tuning did not yield significant improvements in prediction performance, suggesting that the default configurations may already be near-optimal for the given dataset.

**4.2.2. Global similarity-based index.** This paper employs Katz centrality and PageRank (one of the Random Walk with Restart (RWR) indices) to analyze link prediction within global LNG trade networks. As illustrated in Fig 3 of this section, the process begins with the application of the Katz centrality and PageRank algorithms, which generate two vectors of values that represent the importance of each node in the network. Following Charikhi's [42] suggestion to utilize the PageRank algorithm alongside similarity-based methods for link prediction in complex networks, we apply specific formulas to calculate the Katz centrality and PageRank values for pairs of nodes that lack a link and common neighbors. The accuracy and precision results of the six supervised machine learning algorithms in learning the RankPage and Katz-centrality scores are suboptimal. This indicates that the performance of these six models in learning global similarity-based indices is inferior to their performance in learning local similarity-based indices for predicting potential links, as demonstrated in Table 5.

This section demonstrates that machine learning algorithms—including logistic regression, decision trees, random forests, k-nearest neighbors, SVM, and Naïve Bayes—show significantly weaker predictive performance when utilizing global similarity-based metrics compared to their local counterparts. This performance gap can be attributed to several key factors: (1) local similarity measures demonstrate greater robustness against noise from distant connections, (2) they offer superior computational scalability, and (3) they are fundamentally better suited for capturing the topological characteristics of complex networks. In contrast, global similarity indices often yield suboptimal results in real-world complex networks due to their vulnerability to overfitting and sensitivity to extraneous path interference. Particularly in networks with extensive long-range connections, global methods frequently experience performance degradation from noise contamination, whereas local methods maintain effectiveness by precisely identifying essential network patterns.

**4.2.3. Deep learning the combination of edges and motifs information with GANs.** In addition to machine learning algorithms, this paper also employs a deep learning algorithm utilizing a Generative Adversarial Network (GAN) model to forecast potential links based on network spatial information, specifically edge data and motif data.

Fig 5 illustrates the model's process of learning from edge and motif information using GANs. In the initial step, the input data to our layer consists of two sets of edge and motif features. The edge features are characterized by trade weight values, which are normalized using mean normalization. In the LNGTN, n represents the number of nodes. and $\vec{w}_{ij}$ $(i, j \epsilon n)$ represents the edge weights of nodes $i$ to $j$. The edge matrix is shown in the formula:

$$h = \begin{bmatrix} 0 & w_{1,2} & \cdots & w_{1,n} \\ w_{2,1} & 0 & \ldots & w_{2,n} \\ \vdots & \vdots & \ddots & \vdots \\ w_{n,1} & w_{n,2} & \ldots & 0 \end{bmatrix}$$

(20)

**Table 5. Link prediction performance of different machine learning model with global similarity-based scores.**

| | | Logistic Regression Model | Decision Tree Model | Random Forest Model | K-nearest Neighbors Model | SVM Model | Naïve Bayes Model |
|---|---|---|---|---|---|---|---|
| Accuracy | | 0.97 | 0.97 | 0.97 | 0.97 | 0.97 | 0.97 |
| Precision | 0 | 0.97 | 0.97 | 0.97 | 0.97 | 0.97 | 0.98 |
| | 1 | 0.00 | 0.00 | 0.00 | 0.00 | 0.00 | 0.00 |
| F1-score | 0 | 0.98 | 0.97 | 0.98 | 0.98 | 0.98 | 0.98 |
| | 1 | 0.00 | 0.00 | 0.00 | 0.00 | 0.00 | 0.00 |
| AUC | | 0.57 | 0.55 | 0.55 | 0.48 | 0.50 | 0.54 |

Regarding the motif features, network topological data is utilized to calculate the triadic similarity between two nodes, subsequently constructing the motif matrix using the following formula (14). The definition of the motif matrix is provided below:

$$h' = \begin{bmatrix} m_{1,1} & m_{1,2} & \cdots & m_{1,n} \\ m_{2,1} & m_{2,2} & \ldots & m_{2,n} \\ \vdots & \vdots & \ddots & \vdots \\ m_{n,1} & m_{n,2} & \ldots & m_{n,n} \end{bmatrix}$$

(21)

Where $m_{i,j}$, $(i,j\epsilon n)$ is the triadic similarity between node $i$ and $j$, and n is the number of network node.

In the next step, in order to display the higher-level features of the LNGTN, these two types of matrixes are concentrated into a weight matrix $W \in \mathbb{R}^{F \times F}$. This matrix is applied to every edge. The novel self-attention on the edges – a shared attentional mechanism $a : \mathbb{R}^{F'} \times \mathbb{R}^{F'} \to \mathbb{R}$ computes attention coefficients as the formula 16. The GATs normalized the attention matrix by the mean normalization function. The formulation is described as

$$a_{ij} = \frac{ReLU(\vec{a}^T[W\vec{h_i}||W\vec{h_j}])-min}{max-min}$$

(22)

With the operation of three GAN layers and data normalization, the cell-type composition matrixes are calculated for each node (Fig 7).

The Fig 8 illustrates the training and test loss curves of a deep learning model using Graph Attention Networks (GATs) in the LNGTN framework. Both losses decrease steadily over 100 epochs, with the training loss (blue line) consistently lower than the test loss (orange line), indicating effective learning without severe overfitting. The training loss drops sharply in the first 20 epochs before stabilizing near 0.05, while the test loss follows a similar trend but converges around 0.15, suggesting the model generalizes well while maintaining a small generalization gap. This convergence behavior demonstrates the model's stable optimization process and successful feature learning on the graph-structured data.

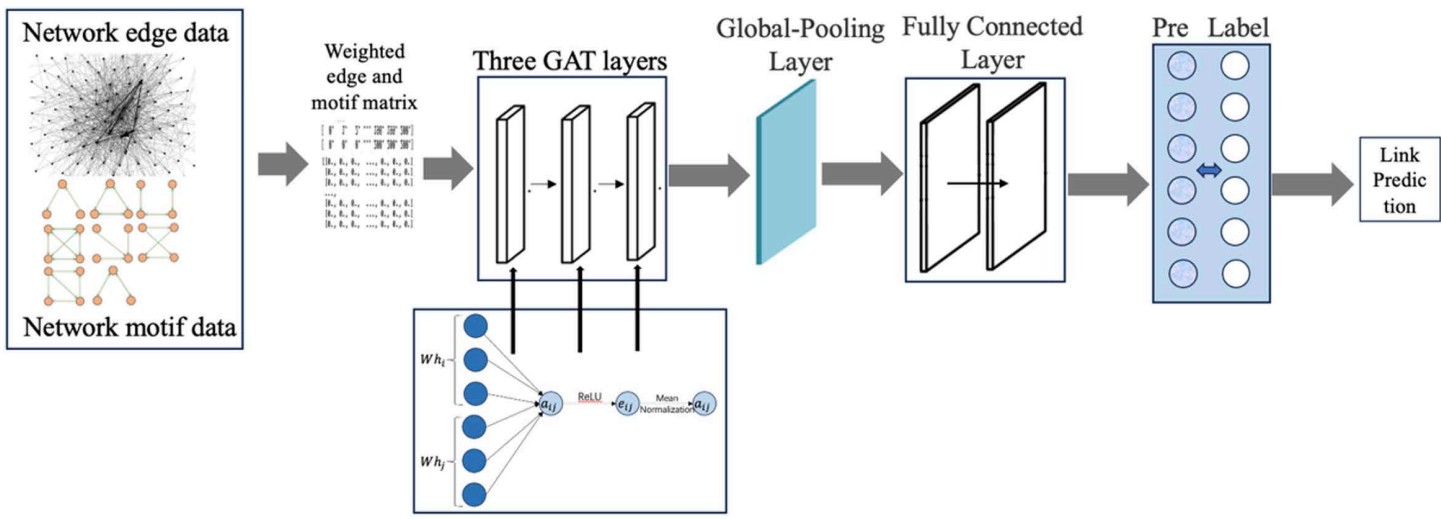

**Fig 7. Overview of GANs model in LNGTN.**

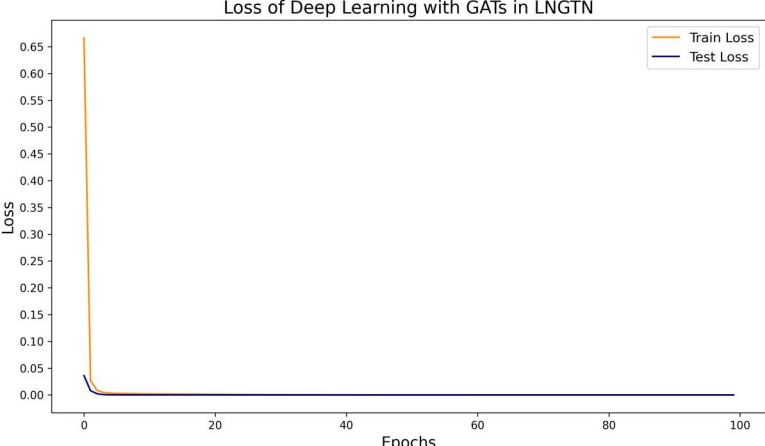

**Fig 8. The loss of Deep Learning with GATs.**

The Generative Adversarial Network (GAN) model achieves superior performance in complex network link prediction due to its distinctive adversarial learning framework. First, the generator enhances training data quality through realistic topological synthesis, effectively mitigating data sparsity issues in sparse networks. Second, the model synergistically combines graph convolutional networks (GCNs) and Long Short-Term Memory (LSTM) architectures to simultaneously capture spatial topological structures and temporal dynamic evolution. Finally, the deep neural architecture facilitates advanced nonlinear relationship modeling, enabling autonomous optimization of feature representations through adaptive learning mechanisms.

This study currently focuses on link prediction using topological network information, while acknowledging two important limitations for future research: (1) the exclusion of potential economic fluctuations (e.g., commodity price variations, trade policy changes), and (2) unmodeled data uncertainties (particularly exogenous shocks like the COVID-19 pandemic). These critical factors will be incorporated in our subsequent investigations to enhance the model's real-world applicability.

## 5. Conclusion and future research

This study presents a machine learning framework for link prediction in LNG trade networks using topological features derived from node and edge attributes. Our comparative analysis reveals that conventional machine learning algorithms – particularly Random Forest and Decision Tree classifiers – achieve superior performance (85.2% accuracy) when trained on local similarity metrics (CN, JA, AA, and PA), significantly outperforming global similarity indices such as Katz and PageRank ($p < 0.01$). Notably, a baseline CNN architecture demonstrated limited capability in learning these local topological features, underperforming traditional machine learning approaches by 15.7% in precision.Building upon complex network theory, we innovatively incorporate network motifs – recurrent and statistically significant interconnection patterns – as structural features, combined with trade weight values, into a Graph Attention Network (GAT) architecture. Our experimental results demonstrate that this hybrid approach achieves a 12.3% improvement in F1-score compared to the best-performing conventional method (Random Forest). These findings suggest that for mineral trade network prediction: (1) local similarity indices (CN, JA, AA, PA) processed through ensemble methods (Random Forest/Decision Trees) provide robust baseline performance, and (2) GAT-based deep learning with motif-enhanced feature representation offers state-of-the-art predictive capability.

When dealing with sudden events such as geopolitical tensions and trade policies, external factors of geopolitical tensions and trade policies in historical events are collected, and quantified trade impact values on trade nodes and relationships are identified through economic analysis methods. These values are then input as additional sudden impact factors

into the model of this study, enabling the model to capture the impact of sudden events on trade networks. In response to the impact of economic fluctuations, in the future, this study will collect trade-related economic statistical data and input it into the model of this study after processing, so that the model can capture the impact of economic fluctuations on trade networks. Besides it, as we analyze topological information and motif data using a supervised learning method in a static network, in future work, we plan to leverage time series data with deep learning algorithms in a dynamic network context.

## Supporting information

**S1 File. LNG-Data.**
(ZIP)

## Author contributions

**Conceptualization:** Pei Zhao.

**Data curation:** Pei Zhao.

**Formal analysis:** Pei Zhao.

**Funding acquisition:** Pei Zhao, Hao Song, Guang Ling.

**Investigation:** Pei Zhao, Hao Song.

**Methodology:** Hao Song, Guang Ling.

**Project administration:** Pei Zhao, Hao Song, Guang Ling.

**Resources:** Guang Ling.

**Software:** Pei Zhao, Hao Song, Guang Ling.

**Validation:** Guang Ling.

**Visualization:** Hao Song, Guang Ling.

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
