## [Decision Letter · Decision Letter 0]

Dear Dr. Song,

Thank you for submitting your manuscript to PLOS ONE. After careful consideration, we feel that it has merit but does not fully meet PLOS ONE’s publication criteria as it currently stands. Therefore, we invite you to submit a revised version of the manuscript that addresses the points raised during the review process.

We look forward to receiving your revised manuscript.

Kind regards,

Gianluca Genovese, Ph.D.

Academic Editor

PLOS ONE

 [Sichuan Oil and Gas Development Research Center 2024 General Projects�2024SY004�.]. 

Reviewers' comments:

Reviewer's Responses to Questions

**Comments to the Author**

1. Is the manuscript technically sound, and do the data support the conclusions?

Reviewer #1: Yes

Reviewer #2: Yes

2. Has the statistical analysis been performed appropriately and rigorously?

Reviewer #1: Yes

Reviewer #2: Yes

3. Have the authors made all data underlying the findings in their manuscript fully available?

Reviewer #1: Yes

Reviewer #2: Yes

4. Is the manuscript presented in an intelligible fashion and written in standard English?

Reviewer #1: Yes

Reviewer #2: Yes

Reviewer #1: 

This study applies machine learning and complex network theory to predict future links in the global liquefied natural gas (LNG) trade network. Overall, the manuscript presents compelling results. However, there are specific areas where additional clarity or elaboration could further strengthen the quality and impact of the work:

1. The conclusion presented in the abstract lacks clarity and fails to effectively summarize the key findings and contributions of the study.

2. Incorporating a literature review table is suggested to clearly highlight the innovations of your research and its contribution to the field.

3. The selection of diverse algorithms such as Random Forest, Decision Tree, SVM, and KNN for comparing their performance in terms of accuracy and predictive capability is very useful. It might be beneficial for the authors to pay more attention to the hyperparameter tuning for each of these algorithms to ensure the use of optimal settings for each model.

4. Although the data for past years has been well presented, it could be interesting to explore how the proposed models would be able to predict new links under future conditions and varying geographical-economic variables.

5. A section discussing the limitations of the proposed models and the challenges ahead (such as data uncertainties or political-economic changes) could enrich the article further.

6. The explanations related to the methodology are somewhat brief. It would be beneficial to provide more detailed and in-depth descriptions of the methods used, including specific steps, assumptions, and rationale behind the chosen approaches. This would enhance the clarity and reproducibility of the research.

7. You may consider adding a "Discussion" section to the paper. This section could provide a deeper analysis of the results, address the limitations of the proposed models, and discuss the implications of the findings in the context of future trade predictions and potential challenges.

Reviewer #2: 

1. Handling Data Sparsity and Imbalance:

Given that the global LNG trade network involves a diverse set of countries with varying levels of trade volume, how does the model handle potential data sparsity or imbalance in the trade relationships? Are there any specific techniques, such as oversampling, undersampling, or the use of synthetic data, that could be implemented to improve model performance in predicting links for countries with minimal trade interactions?

2. Generalizability of the Model.

While the study focuses on the LNG trade network, how do the authors assess the generalizability of the proposed method to other global trade networks, such as oil, minerals, or renewable energy? Are there any adjustments or considerations that need to be made for the model to apply to other commodities with different trade dynamics and structures?

3. Impact of External Factors on Model Performance.

In the paper, the authors primarily focus on network topological features for link prediction. However, external factors, such as geopolitical tensions, trade policies, and economic fluctuations, can significantly affect trade relationships. How might the authors incorporate these external factors into the model to improve its robustness and accuracy, particularly during periods of geopolitical instability or economic crises?

4. Data availability.

It is recommended to upload and publicly share the entire dataset used in this article, rather than simply providing the name of the database.

---

## [Decision Letter · Decision Letter 1]

Machine learning approaches for predicting the link of the global trade network of Liquefied Natural Gas

PONE-D-25-15370R1

Dear Dr. Song,

We’re pleased to inform you that your manuscript has been judged scientifically suitable for publication and will be formally accepted for publication once it meets all outstanding technical requirements.

Kind regards,

Gianluca Genovese, Ph.D.

Academic Editor

PLOS ONE

**Comments to the Author**

Reviewer #1: All comments have been addressed

Reviewer #2: All comments have been addressed

2. Is the manuscript technically sound, and do the data support the conclusions?

Reviewer #1: (No Response)

Reviewer #2: Yes

3. Has the statistical analysis been performed appropriately and rigorously?

Reviewer #1: (No Response)

Reviewer #2: Yes

4. Have the authors made all data underlying the findings in their manuscript fully available?

Reviewer #1: (No Response)

Reviewer #2: No

5. Is the manuscript presented in an intelligible fashion and written in standard English?

Reviewer #1: (No Response)

Reviewer #2: Yes

Reviewer #1: I would like to express my sincere appreciation to the authors for their thoughtful and thorough responses to my comments, as well as for the revisions made to improve the quality of the manuscript.

Reviewer #2: (No Response)

**Do you want your identity to be public for this peer review?** For information about this choice, including consent withdrawal, please see our Privacy Policy

Reviewer #1: No

Reviewer #2: **Yes: ** Po-Yin Chang

---

## [Editor Report · Acceptance letter]

PONE-D-25-15370R1

PLOS ONE

Dear Dr. Song,

I'm pleased to inform you that your manuscript has been deemed suitable for publication in PLOS ONE. Congratulations! Your manuscript is now being handed over to our production team.

Kind regards,

on behalf of

Dr. Gianluca Genovese

Academic Editor

PLOS ONE